# From Sea to Table: The Role of Traceability in Italian Seafood Consumption

**DOI:** 10.3390/foods14203469

**Published:** 2025-10-11

**Authors:** Simona Violino, Damianos Chatzievangelou, Giulio Sperandio, Simone Gaetano Amato, Chiara Fini, Domenico Ciorciaro, Simone Figorilli, Cecilia Ripa, Simone Vasta, Francesca Antonucci, Federico Pallottino, Raffaele De Luca, Daniela Scutaru, Sara Biancardi, Elisa Pignoni, Corrado Costa

**Affiliations:** 1Consiglio per la Ricerca in Agricoltura e L’analisi Dell’economia Agraria (CREA)—Centro di Ricerca Ingegneria e Trasformazioni Agroalimentari, Via della Pascolare 16, Monterotondo, 00015 Rome, Italy; giulio.sperandio@crea.gov.it (G.S.); simonegaetanoamato@gmail.com (S.G.A.); chiara.fini50@gmail.com (C.F.); simone.figorilli@crea.gov.it (S.F.); cecilia.ripa@crea.gov.it (C.R.); simone.vasta@crea.gov.it (S.V.); francesca.antonucci@crea.gov.it (F.A.); federico.pallottino@crea.gov.it (F.P.); daniela.scutaru@crea.gov.it (D.S.); sara.biancardi2@gmail.com (S.B.); corrado.costa@crea.gov.it (C.C.); 2Instituto de Ciencias del Mar (ICM-CSIC), Consejo Superior de Investigaciones Científicas, 08003 Barcelona, Spain; damianos@icm.csic.es; 3Department of Science and Technology (DiST), Marine Ecology Laboratory, Parthenope University of Naples, Centro Direzionale—Isola C4, 80143 Napoli, Italy; domenico.ciorciaro001@studenti.uniparthenope.it; 4Fondazione Massimo Spagnolo E.T.S., Via S. Leonardo 81, 84131 Salerno, Italy; delucaconsulenze@gmail.com; 5Department of Environmental and Prevention Sciences, University of Ferrara, Via Saragat 1, 44122 Ferrara, Italy; elisa.pignoni@unife.it

**Keywords:** willingness to pay, online market, made in Italy, fresh seafood, consumers’ preferences, supply-chain, economic sustainability, blockchain

## Abstract

Seafood plays a key role in a healthy diet due to its high content of essential nutrients. However, its global trade and complex supply chains expose it to frequent mislabeling and food fraud. This study investigates Italian consumers’ willingness to pay (WTP) for traceable seafood products, exploring how blockchain technology (BT) can enhance transparency and economic sustainability in the fish supply chain. An online questionnaire, administered in 2022 and 2024, gathered responses from a diverse demographic, focusing on four representative seafood species: farmed sea bass, striped venus clams, giant red shrimp, and albacore tuna. Results indicate that most respondents—primarily with higher education levels—value traceability and are willing to pay a premium for certified, traceable seafood. The study models the economic feasibility of implementing BT at both national and regional levels and finds that the consumer’s WTP exceeds the additional costs incurred by adopting BT. These findings support the viability of traceability systems in improving food safety and sustainability, while reinforcing consumer trust. The results also underscore the importance of providing clear information at the point of purchase, particularly regarding species, origin, and production methods—factors critical to informed seafood choices and advancing more sustainable consumer behavior in Italy.

## 1. Introduction

### 1.1. Seafood Consumption and Safety Caveats

Seafood products contribute to a healthy and balanced nutritional diet, due to their composition of key macro- and micro-nutrients, including omega-3 fatty acids and vitamins, contributing to a healthy and nutritionally balanced diet [1].

Accordingly, its consumption has been historically integral for communities settled near coastlines, with indications of opportunistic gathering of shellfish dating back to the Mesolithic period [2,3]. In recent years, improvements in logistics, including transport, storage, and preservation technologies, allowed fish products to become the most traded food in the world in terms of quantity, far surpassing coffee, sugar, wheat, and rice among others [4].

Global fish consumption per capita has increased from 9 kg to 20.5 kg (122%) between 1990 and 2018 [5]. The European Union is the world’s largest importer, with more than half of seafood originating from Norway, the UK, China, Morocco, and Ecuador, where fishing supports many local communities. Italy covers about 8000 km of coastline (fifth longest in Europe), and the main landing coasts include Sicily, Marche, Veneto, the southern Tyrrhenian coast, and Puglia. However, the greatest demand for seafood products is related to fish farms (trout, sea bass, sea bream, mussels and clams), instead of direct landings [6]. In 2019, the average per capita consumption by Italians, according to data compiled by the association SOS Italian Fish, was about 28 kg of fish per year, higher than the European average. Nevertheless, this figure is decidedly low when compared to that of other European nations with a similar coastline, such as Portugal (60 kg); [7]. Two-thirds of the national demand for fish is met from the rest of the oceans, particularly from developing countries [5]. A study conducted by Unioncamere Puglia, shows that 57.1% percent of Italians consume fish at least once a week, especially younger people [8]. In addition, Italians’ consumption of fresh fish accounts for nearly a quarter of the total in the 12 countries belonging to the European Union (Germany, Denmark, Spain, France, Hungary, Ireland, Italy, the Netherlands, Poland, Portugal, Sweden and the United Kingdom), and from 2018 to 2019 it increased by 2% in both volume and value, with a five-year peak in the latter of 3.21 million [9].

Blockchain technology (BT) is revolutionizing traceability in the food industry, with an increasing emphasis on transparency and trust. An innovative approach has been employed by Bofrost Italia, which has adopted BT with Nordic Cod Fillets and Artichoke Heart Wedges through the EY OpsChain Traceability platform, ensuring greater product safety and integrity within the frozen food supply chain. By recording information such as fishing methods, MSC certification for sustainable fishing, freezing processes, packaging, storage conditions, and quality control, this technology is a powerful means of ensuring safety and accountability within supply chains [10].

However, advances in the seafood supply chain have not completely eliminated food safety risks associated with the processes from harvesting the fresh product to it reaching the final consumer [4]. Specifically, food risks vary depending on the region, local environmental conditions, management practices, and methods of production, preparation, and consumption of seafood products. The greatest risks can be caused by parasitic infections, diseases induced by pathogenic bacteria, and heavy metal contamination [11].

The Food and Agriculture Organization of the United Nations [12] defined food safety as “the style of preparing, handling, and storing food to prevent infection and to help ensure that food retains sufficient nutrients for a healthy diet. Unsafe food means that it has been exposed to pathogens, or rotten agents, that can cause illness or infection (e.g., diarrhea, meningitis, etc.)”. In this regard, European regulation 91/493/EEC [13] delineates the general rules for the handling, production and placing on the market of fish products for consumption, specifying the legislation governing food hygiene and established specific safety standards for food of animal origin (including fish products). Regulation No. 178/2002 [14] establishes the general principles and requirements of food law, provides procedures on food safety, and deals with the basic (fundamental) concepts of equivocality and traceability. In 2004, Regulation No. 853/2004 [15] of the European Parliament and Council established the specific health rules for food of animal origin (acceptance of establishments, health and identification marking, imports), including aquatic animals.

These aspects highlight the importance of quality control for the governing bodies, which is in turn a reflection of its importance for the citizens/consumers.

### 1.2. Importance of Seafood Traceability and Its Reflection on Consumers’ Behavior

Geographical traceability plays a central role, as it allows the legality and sustainability of fishing practices to be demonstrated, correcting the information asymmetry that characterizes the sector [16]. For example, the European Union and the United States, the main consumer markets, have introduced stringent regulations on the traceability of fish products. Article 58 of Regulation (EC) No. 1224/2009 [17] stipulates that all batches of fishery and aquaculture products must be traceable throughout the entire supply chain, from catch to retail sale. more recently, Regulation (EU) 2023/2842 [18] has strengthened these requirements by introducing digital systems and extending them to imported products [4,5,6,7,8,9,10,11,12,13,14,15,16,17,18,19]. Several studies in the literature confirm the importance of traceability in consumer behavior. In Italy, a study found that consumers are willing to pay a premium of +4.75% to know the catch area of processed fish products such as seafood salads and marinated anchovies [20].

In emerging markets, such as Bangladesh, the attributes that most influence purchasing behavior are the production method and safety guarantees (e.g., “formalin-free”), with a clear willingness to pay a higher price for products with traceability information [21]. At the European level, Menozzi et al. [22] highlighted a willingness to pay for nutritional and sustainability claims: in Italy, for example, the average premium for health claims is €0.96/kg, while for sustainability it varies according to species (e.g., herring €2.93/kg).

In addition to traceability, other subjective factors also influence the perception of the quality of fish products: brand and price are often considered important indicators, especially when clear information or a recognizable brand is lacking, as is frequently the case with fresh fish sold at the counter. Although consumers say that quality is the most important variable for them, the literature shows that they also pay attention to additional labeling on packaging. In addition, purchasing decisions are influenced by factors such as perceived health benefits and risks, the sustainability of the production method, geographical origin, and convenience of use [23].

An international survey conducted by the Marine Stewardship Council reveals that 55% of consumers doubt the accuracy of labels and that 65% want a traceable and reliable supply chain, with the ecolabel perceived as a sign of trust [24].

Therefore, traceability is not only a technical mechanism for fisheries management, but also a theoretically sound tool for mitigating market failure linked to imperfect information, with direct implications for consumer behavior and willingness to pay.

Traceability applied to seafood products allows customers to obtain reliable information about the products they purchase, ensuring certainty of product quality [25]. Additionally, it helps them understand the higher price of a quality product, so that producers can sell a sustainable, non-imported, certified product of documented freshness [26]. Therefore, robust traceability systems enhance international trade of fish products, demonstrating the consistency and transparency of regulations and reducing the likelihood of possible fraud in this area [11]. Finally, traceability of fishery products can promote sustainable fisheries management by demonstrating that a particular product has been legally fished [4], a pressing issue on a global scale [27]. As such, traceability affects consumers’ preference towards a specific seafood product, with the country of origin (either domestically caught or locally raised over imported), as well as sustainability (origin and ecolabeling) being common factors [28].

Consumers’ “willingness to pay” (WTP) as a result of the importance of traceability represents a fundamental marketing strategy for the price-response models that inform optimal decisions about prices and promotions. Through this information, companies are informed about how much consumers are WTP early in the product development process, and researchers quantify the value of a product [29]. Overall, WTP premiums for traceable food products enable agribusiness producers and retailers to invest in differentiating their products in the marketplace, to command higher prices and to build brand loyalty and a reputation for quality and responsibility, ultimately leading to increased sales and profitability in the long run [30]. Violino et al. [31] used an online questionnaire to analyze Italian consumers’ overall interest and WTP for food products with guaranteed traceability, and report that the majority view traceability as a key aspect of extra virgin olive oil (EVOO).

Unfortunately, seafood products are hard to identify and trace and, being the most traded food products in the world, they are also the most prone to labeling errors and international frauds. In fact, in cases of highly diverse faunal groups consisting of multiple species of similar morphological characteristics, removing external features during processing steps could make morphological identification nearly impossible, often leading to the fraudulent substitution of a high-quality seafood product with one of lower cost [32]. All these make traceability throughout the supply chain more complicated than the straightforward “one up and one down” traceability, with species of origin (fish species), production method (wild or farmed, organic or intensive) and geographic origin (fish from different regions) being particularly challenging [33]. Current methods for determining the origin of seafood products include various analytical techniques [1], including:DNA profiling-based techniques, blockchain technologies, and Radio Frequency Identification (RFID) systems to identify speciesTechniques based on DNA profiling, fatty acids, stable isotopes, blockchain, RFID, X-ray fluorescence (XRF) through Itrax, and Inductively Coupled Plasma-Mass Spectrometry (ICP-MS) to determine production methodsDNA profiling, fatty acid profiling, stable isotope profiling, blockchain, RFID, X-ray fluorescence (XRF) through Itrax, and Inductively Coupled Plasma-Mass Spectrometry (ICP-MS) techniques to determine geographic origin.

Finally, software-based densitometry and image analysis allow us to overcome subjective evaluation of patterns, making it possible to correctly identify already filleted or sliced flatfish or gadoid fish [34].

Given the importance of traceability of products to consumers (subjective), the potential health implications (objective), and the related technological/analytical requirements, the availability of information on product quality is essential, especially at the time of purchasing [21].

This study assesses Italian consumers’ WTP defined as a premium on top of the conventional selling price for a certified seafood of certain origin, using an online questionnaire that focuses on four fish species (i.e., farmed sea bass, striped venus clams, giant red shrimp, and albacore tuna, all identified as representative of a wider range of seafood products). The final objective is to assess the economic sustainability of the implementation of a traceability system for fish supply chain products (caught and/or farmed products) by applying an advanced system based on blockchain technology (BT), applied with reference to different scenarios on national and regional scales.

Despite the growing attention paid to blockchain in agri-food traceability, the literature shows a lack of systematic assessments focused on the Italian fishing industry. No previous study has jointly analyzed implementation costs and consumer willingness to pay to assess the economic sustainability of blockchain in fishing. This study fills this gap by providing empirical evidence linking technological feasibility to consumer behavior.

## 2. Materials and Methods

### 2.1. The Questionnaire

To assess consumers’ attention to seafood traceability, a questionnaire in Italian was launched on Microsoft Forms on two different occasions and remained active for 70 (1 September to 9 November 2022) and 228 (17 June 2024 to 31 January 2025) days, respectively. The complete, translated questionnaire is provided in Table 1. In the first phase, the main dissemination channels were the website and social media accounts of the Council for Agricultural Research and Analysis of Agricultural Economics (CREA; www.crea.gov.it/home (accessed on 10 September 2025); www.facebook.com/CREARicerca (accessed on 10 September 2025); www.instagram.com/crearicerca (accessed on 10 September 2025)), as well as promotion through Instagram, Facebook and WhatsApp stories by the manuscript’s authors. The second phase also included a leaflet campaign. In addition, CREA’s mailing list and flyers with a QR code on the fish counters of Coop Centroitalia GDOs were used to collect rapid responses and to raise consumer awareness when choosing a fish product.

The survey primarily aimed to detect consumers’ interest and to receive additional information on consumption and traceability of seafood products. Based on this preliminary information, consumers were asked to give their opinion on whether they would be WTP an additional cost above the conventional selling price for this service. The questionnaire was divided into two sections: sociodemographic questions, to identify the characteristics of the respondents, and specific questions related to the research objective. The sociodemographic questions covered gender, age, range, geographic area of residence, and education level. Within the questionnaire, a short video in Italian (https://www.youtube.com/watch?v=j84DAlQNMbw (accessed on 10 September 2025)) was presented in order to explain what traceability is and why it is important. Specific questions covered: whether and how frequently seafood was consumed and how often seafood was consumed, consumer interest in product origin and interest in seafood traceability, and WTP (additional cost) for four particular types of seafood products identified as most representative of a broader landscape.

Sea bass (*Dicentrarchus labrax)* [35] reared in Italy using extensive or intensive techniques (i.e., lower or higher control over the production factors, respectively), is one of the most commercially popular fish species. Italian farms ensure greater freshness and better organoleptic characteristics due to the quality of the food provided.

Striped venus clams (*Chamelea gallina*) [35], are medium-sized clams typical of the Adriatic Sea. They have a striped, clear shell and very tasty, flavorful meat and excellent nutritional value. They are marketed in nylon nets.

The giant red shrimp (*Aristaeomorpha foliacea*) [36] lives mainly in muddy bottoms between 400 and 1300 m depth. Unable to be sold fresh, the temperature is lowered on board by an instant freezing process. It is one of the most valuable fish products and it is prized for the tastiness of its meat.

Albacore tuna (*Thunnus alalunga)* [37] is caught and processed fresh. The slices are cooked in brine with herbs and preserved in glass or tin jars with salt and extra virgin olive oil.

These four fish species were selected because they represent different types of edible seafood and cover a wide range of commercial value (from 10 to 50 €/kg). In fact, sea bass was chosen as representative of fish sold in full; striped venus clam represent a seafood product sold in nets; red shrimp is a valuable product sold in trays; and, finally, albacore tuna is representative of processed and preserved seafood.

It has to be noted that by default the completion of the specific questions (i.e., after Q5 of the questionnaire) does not make sense unless the participants consume seafood (Q5 reply “Yes”).

### 2.2. Methodological Remarks

The online questionnaire was selected as an assessment method for consumer behavior, following Violino et al. [31]. In total 4058 participants responded, which might be considered a small sample when compared to a national population of 59 million. However, this figure is not too dissimilar to the normal participation in opinion polls for national elections, and we are confident of the representativeness of the sample in terms of size. It has to be kept in mind that our approach, being an open questionnaire, is not equivalent to fixed, pre-determined sample-sized medical treatment experiments, for instance.

Not following a stratified approach can create underrepresentation of certain demographics or other groups, leading to potential discrepancies when pooling the outcomes of the survey. The results should therefore be interpreted as rather exploratory, instead of representative of an entire nation. However, we took measures to minimize this potential bias and increase the confidence of the results, which we list below.

Some concerns could be raised by the fact that the dissemination media could skew the profile of the participants: (a) mostly online and (b) through research institutes’ channels. This is a potential limitation we considered at the time of designing the study and is reflected in the under-sampling of 66+ year old age group and the disproportional representation of BSc (or higher) degree holders. Our effort to reach the public through the supermarkets counterbalances this to some extent. Also, to tackle this, we do not compare absolute frequencies across groups of different sizes, but ratios within them. To this end, we adapted the analysis to a Poisson sampling scheme (i.e., open sample size, row and column totals), instead of, for instance, joint multinomial sampling (i.e., only sample size is fixed), independent multinomial sampling (i.e., sample size and one of row or column totals are fixed), or hypergeometric sampling (i.e., sample size and both row and column totals are fixed). That decision of sampling scheme affects the way Bayes Factors (see next section) are calculated, and thus, the reported significance [38]. At the same time, following a Bayesian approach based on simulating the data 10^4^ times helped smooth the effect of outliers. It compares the observed ratios to the expected ones based on the distribution of groups within the general population.

Finally, it goes without saying that an online questionnaire about seafood consumption will attract more seafood consumers than non-consumers. Thus, we do not intend to represent the entire Italian population, but the part of it that consumes seafood (or at least is not against it in principle) and is more willing or prone to participating in online surveys. As such, the results represent an exploratory framework.

### 2.3. Statistical Analysis

A preliminary descriptive statistic was conducted observing socio-demographic categories in relation with consumers’ attitudes.

For questions with categorical outcomes (i.e., all but the 4 questions on extra prices), participant counts per answer were treated as Poisson events [39] that can be described by a discrete probability distribution, with an expected rate λ ∈ (0, ∞). Plainly, the observed n frequencies are compared against the expected frequencies λ (based on the proportion of each group within the total sample), with possible statistical differences inferred by low probabilities.

A Bayesian approach was followed to provide full distributions for the estimations of the λ parameter, while at the same time allowing for the quantification of the support in favor of the null hypotheses (i.e., that observed frequencies of the response variable groups are equal to the frequencies of the predictor groups), instead of only against them [40].

Bayes Factors were used as analogues of the classic contingency table analysis for count ratio comparisons. In particular, the independence assumption in a contingency table under a Poisson sampling plan without a fixed total size N [38] was tested with the package “BayesFactor” [41] of the R Statistical Language [42]. Probability density distributions for each λ (i.e., for each unique combination of predictor and response) were estimated from 10^4^ posterior draws generated by a Markov Chain Monte Carlo [43] sampling process (i.e., simulation of a parameter in sequential steps, with the value of step i depending only on the value at step i − 1, independently of the initial step 0 and the process that led to i − 1). Density plots were used to visualize the distributions of all λ rates.

For the 4 questions on extra prices (i.e., with numeric outcomes), a Bayesian Analysis of Variance (ANOVA)-like comparative method based on linear mixed models [44] was used (R package “BayesFactor”) to derive potential statistical differences among different groups of participants, being robust to assumptions of deviations from normality and homogeneity of variance. The probability density distributions from 10^4^ posterior MCMC draws were visualized with density plots.

All analyses were performed both for the 2022 and 2024 subsets individually, as well as for the full dataset.

### 2.4. The Economic Analysis

#### 2.4.1. Description of the Economic Model

The economic analysis is based on the net present value (NPV) method, widely used in agri-food economics to assess the feasibility of investments [45]. In this study, the consumer WTP premium, obtained through the survey, was used as a proxy for economic benefit, in line with previous applications of blockchain adoption in food systems [46,47]. This theoretical framework allows for a rigorous comparison between implementation costs and the added value perceived by consumers.

The aim of the economic analysis was to investigate the possible economic sustainability of the application of new traceability technologies for fish products, represented by implementation of the BT for the fish supply chains covered by the questionnaire. This was done to calculate, for each supply chain, economic sustainability by highlighting the break-even point calculated by considering the impact of the new product traceability technologies compared to a traditional traceability system. The economic sustainability judgment was expressed based on the consumer’s WTP a price higher than the average market price to purchase a product that is completely traced and certified from the origin. This “premium price” was obtained, for each supply chain, from the results of the anonymous survey conducted among final consumers, which highlighted the percentage increase in price, compared to the standard of each product, that the consumer himself is WTP to have a product traced in a certain and unequivocal way and certified from the moment of its capture at the sales counter.

The scenarios analyzed, with reference to the market price reported in the questionnaire, were therefore the following:(1)National farmed sea bass supply chain (BS) (18 €/kg);(2)Supply chain of clams (CS) in bags (10 €/kg);(3)Supply chain of quality red shrimp (RS) in cans (50 €/kg);(4)Supply chain of white tuna processing (TS) (40 €/kg).

From an economic point of view, the analysis relating to the application of BT in fish supply chains essentially pursued two purposes: the first is linked to the definition of a model for evaluating the costs of applying new technologies in relation to the four identifying scenarios of the supply chain; the second is to identify the limit of economic convenience of the traceability process with BT considering both the level of initial technological investment required and the subsequent annual and periodic maintenance and management interventions to guarantee the operational efficiency of the technological platform that should produce potential benefits to end users in terms of certain identification of the product, useful information for health protection, as well as improvement and optimization of one or more phases of the production process.

The economic evaluation methodology of the application of BT also considers what has been proposed by various authors [48,49,50,51,52] for other supply chains in terms of product traceability, The analysis developed is essentially divided into the following phases:
(a)Implementation of all the components necessary to set up the technological traceability system based on the BT with determination of the initial capital to be invested, development of the necessary software and hardware, as well as the cost of training suitable personnel for the start-up and management of the system;(b)Installation of the system and its use with definition of all the operations necessary for its regular functioning; detection of current costs to proceed with its application along all the points of increase of the information provided to the system along the production chain: farming or landing phase; product transport phase; packaging and processing phase; distribution and marketing phase. The electronic data collection and recording system for product traceability and monitoring must be applied to each essential step with transfer of the required information with the addition of new information relevant to the specific level reached along the production chain.(c)Implementation and modernization of the systems for receiving and reading the information stored at the level of individual farms (farmed fish) or vessels (caught fish) and, by gravity, on the quantities of fish unloaded on land and transported with various passages to the final destination;(d)Once these investments have been addressed and well defined, it is then necessary to verify the benefits that BT can generate within the production chain. The economic benefit can be assessed in relation to the improved efficiency in carrying out the operations of identifying the quantity and quality of the fish product;(e)The last phase is that of the overall analysis of the convenience of using the BT, to be assessed on a small scale, referring to case studies of small local/regional realities, and on a large scale, for example national, where the widespread application of the new traceability technologies can also generate possible benefits to be attributed to any economies of scale achieved.

On the basis of the development of the points listed above, an evaluation model was formulated to express a judgment of convenience on the application of BT, identifying, as previously mentioned, the levels of potential economic break-even (break-even point), calculated on the basis of the final consumer’s WTP a higher price for a “technologically traced” commercial product compared to the current conditions of uncertain traceability of the product. Currently, the costs attributable to the application of this legislation aimed at identifying and qualifying the product in the various phases of the supply chain are essentially identifiable with the costs of labor and equipment necessary to fulfil the legislative obligations to ensure transparency and traceability of production processes along the supply chain.

#### 2.4.2. Case Studies Examined

In a BT implementation system, all the required information must be digitally entered into the technological platform that will record the individual transactions of quantitative/qualitative data with the relative time track of the event. The overall costs, instead, refer to all the possible implementation costs of the technological system based on the BT that, in this study, have been determined considering a small- or large-scale BT application. Based on the dimensional level of BT implementation, it is possible to identify four main cost items:Development of the BT implementation software that requires highly qualified professionals in the IT sector;Availability and use of appropriate hardware with high storage memory capacity;Creation/programming of the BT and its management and update;Appropriate training of specialized personnel for the management/maintenance of BT.

Based on the elements reported above, an estimate of the implementation costs of a BT for each scenario has been evaluated. For the four supply chains, the implementation of the BT was evaluated on large (L) and small (S) scale: the first refers to fish production on a national scale; the second, instead, refers to more limited productions attributable to more circumscribed local realities and compatible with regional production. In the latter case, reference was made to an estimate of fish production equal to approximately 6.7% of national production, considering that the largest contribution to national fish production comes from 15 Italian Regions that fall within the scope of the European Maritime, Fisheries and Aquaculture Fund [53]. Table 2 shows the national quantities produced [54] and the estimated regional quantities per single supply chain and the variation in prices used subsequently to develop a sensitivity analysis.

Table 3, instead, shows the implementation of a blockchain system for the traceability and certification of the products along the supply chains, developed on a small and large scale. The costs are represented by initial investment costs and the annual and periodic management and updating costs necessary for the efficient maintenance of BT. The economic–financial assessment considers an estimated duration of the investment in BT equal to 15 years, with a system update frequency, considering the speed of technological progress, to be carried out every 5 years (two interventions in the period considered). Table 3 was compiled based on previous experience within the PESCA-CHAIN PROJECT (a project studying the economic and technological feasibility of introducing innovative traceability technologies in the fishing industry) and with the help of the AWS pricing calculator (https://calculator.aws/#/createCalculator/ManagedBlockchain (accessed on 10 September 2025)) for 2024. The economic estimate of BT was also performed based on the work of Violino et al. [31].

#### 2.4.3. Economic Analysis of the Application Scenarios of BT

The costs related to the application of BT generally concern the purchase and implementation of the technology, the development of the software, the hardware and the cost of current management of the system and training of the workers. The impact of the application of BT technology varies significantly depending on the size of the reference system and the amount of data to be processed. It should be considered, in fact, that, at least as regards the definition of BT architecture and the development of the software, the cost varies little in reference to the extension of the territorial application level. In the evaluation of economic sustainability, with the same basic cost of setting up the system, the cases of possible greater sustainability were directly dependent on the greater quantity of product processed in the year. The assessment aimed to establish the possibility of reaching the break-even point in relation to the hypotheses of possible benefits produced by the application of the technology with respect to both the cost attributable to the current mandatory system of traceability of the fish product, and in reference to a price increase tolerated by the final consumer in exchange for secure information on the origin and traceability of the product with the BT.

The main economic indicator considered in the investment analysis was calculated by adopting the Net Present Value (NPV) method, but with reference to the financial analysis of only the investment and management costs of the implementation of the BT, from which to derive the percentage incidence of this cost with respect to the market price of the specific supply chain product. This allows for a direct comparison between this cost increase and the percentage increase in the market price, derived from the survey, that the end consumer was WTP to have a product certified by BT. The time frame for which the investment analysis was carried out, applying the discounting method of all costs attributable to the construction of BT, was 15 years, a reasonable period as amortization time for this technology, while the discounting rate applied was considered equal to 3.56%, as established by the MISE for the year 2023 [55].

## 3. Results and Discussions

It is important to emphasize that the non-probabilistic nature of an open online survey does not guarantee representativeness of the general population, even with a large sample size. This is not expected to have a major effect on contingency table and ANOVA analyses, as they acknowledge the differences in group sizes. However, it can limit the formalization of confidence intervals, and the generalization of the results to a nationwide level. Furthermore, it can potentially overestimate the NPV outcome, under the assumption of the higher proportion of higher educational levels in the survey sample corresponds to an indirect overrepresentation of higher incomes, leading to greater WTPs. However, the Bayesian ANOVA showed that studies affected the WTP premium in less than half of the cases (see the results below and the Appendix A for details), and in many of these the effect was negative (i.e., lower educational levels corresponded to greater WTP). Therefore, pinpointing the exact nature and magnitude of this effect is more complicated, and would require further research.

### 3.1. Socio-Demographic Composition and Effects

A total of N = 4058 respondents participated in the survey: 1265 in 2022 and 2793 in 2024. 52% of the respondents were females; considering the geographical residence area, 33% lived in Central, 28% in the South, while 39% lived in the North of Italy. As for the composition by age group, this was broken down as follows: 13.3% of respondents were 25 years old or under and 13.1% were over 66 years old, 28.7% were between 26 and 40 years old, and 45% were between 41 and 65 years old. Regarding the breakdown by age group, the percentage of the two classes up to 40 years of age are comparable with the Italian National Statistics Institute [56] demographic values of the Italian population, while there was an under-sampling of the population over 66 years of age in favor of the 41–65-year-old class (as shown in Figure 1).

Respondents were divided into three levels of education. Specifically, 41.8% held a bachelor’s degree or higher (e.g., Ph.D., master’s degree, etc.), 39.2% of respondents had a high school education, and only 19% had an inferior middle license. The result on educational attainment is clearly in contrast (significantly higher percentage of college graduates than non-graduates) to the educational attainment estimated by ISTAT et al. [56], Figure 2.

### 3.2. General Seafood Consumption and Preferences

Regarding the seafood consumption questions, in 2022 only 5.1% (n = 64) of the respondents declared they do not consume seafood products; this percentage increased in 2024 (9%; n = 255), (Figure 3).

Considering the relationship between the geographic area where the respondent mainly live and the place where they buy seafood products, it could be observed that, in 2022, consumers in the southern regions tended to purchase seafood from the fish market 49%), while those in the central and northern regions preferred large retailers (57% for the center, 63% for the north). This result was confirmed in 2024, where fishmongers and local markets were predominantly frequented by people from southern Italy (local market 60%; fishmongers 51%), (Table 4).

Between the two periods no differences in the type of product consumed have been observed. Overall, Italian consumers consumed mainly fresh fish products (63%), followed by frozen (31%). Only 6% consumed processed fish products (Figure 4).

Consumers residing in Southern Italy preferred fresh products (57%) compared to those in the North (43%), Figure 5.

It can be observed that, for 2024, the level of information about the origin of seafood products increased with increasing age (for the age group of 66 years old, 92%; 41–65 (79%); 26–40 (66%); and 60% for the youngest group). This relationship was also observed by Fiorile et al. [57] and is further confirmed in the study by Myae & Goddard [58]. Traceability seems to be more important for adults, who are more careful about the safety, freshness, and quality of the products they buy. Moreover, the data regarding the actual interest in traceability also reflect this trend (86% for the youngest group; 92% for the 26–40 years old group; 94% for the 41–65 years old group and 99% for the 66 years old group). Young people, in fact, are the least informed (60%), followed gradually by the other age groups (66% for the 26–40 years old group and 79% for the 41–65 years old group) with people over 66 years old being the most informed (92%). This work is striking because of the data collected for 2024. In fact, 99% of respondents over 66 years old show themselves to be aware of what they buy.

The place of purchase seems to be an important factor regarding information. In 2024 the places where information is greatest seem to be fishmongers and local markets for the age groups of 66 years old, that is, the most informed group, and which prefer local markets (42%) and fishmongers (20% that is the highest percentage respect to the other groups) although the purchase by large retailers cover the 35%. Here, provenance information is provided to the buyer either through labels (84%) or through the retailer (14%). The market influenced the way consumers have access to information on origin. Consumers buying in the fish market and in the local market trusted the info provided by the retailer (31% and 33%, respectively) more than those buying from large retailers (4%) that are informed mainly by the product labels (91%) Similarly, groups buying fresh seafood were comparatively more likely to be informed directly by the retailer (26%) relative to the groups buying frozen (3%) and processed (4%), although the main source of information was still product labels, with the highest percentage for processed products (92%), 90% for frozen products, and 69% for fresh seafood. The type of food purchased also plays an important role in provenance information. In fact, fresh fish consumers state that they are the most informed about “traceability” (83%), followed by consumers of frozen products (61%) and finally processed products (57%). Furthermore, the purchase of fresh, frozen and/or processed products also plays an important role in the frequency of consumption.

Those who buy fresh products tend to consume fish more frequently (31% consume fish 6–10 times per month and 22% more than 10 times per month) than those who prefer to buy frozen (10%, 6–10 times per month and 12%, more than 10 times per month) and processed (34% for 6–10 times per month and 10% more than 10 times per month). It is interesting to note that as in 2022 the interest in the origin of fish products differed according to the type of purchase. In general people who consumed transformed seafood are not really interested in the traceability of the product; in fact, 45% of people declared that traceability is not important. Otherwise, 89% of consumers of fresh seafood and the 62% of frozen products declared that they are informed about the provenance of the product. Another important aspect concerns the relation between the characteristics of the product and the geographical origin.

In fact, although consumers from all around the country generally prefer fresh products and tend to avoid processed ones, the rate of buying frozen seafood increased moving from the South towards the North both in 2022 and in 2024 (15% for the South and 37% for the North in 2022; 22% for the South and 47% for the North in 2024). The same trend was observed for processed products in both years (12% for the South and 56% for the North in 2022; 17% for the South and 53% for the North in 2024), as reported in Table 5.

In general, considering both years, the frequency of consumption that constituted the highest percentage was the range of 1–5 days per month (52%), followed by the range of 6–10 (32%) and more than 10 (16%), (Figure 6).

People who came from the North of Italy were occasional consumers of seafood, preferring to eat it 1–5 times monthly (60.3%), in contrast with people who came from the South of Italy, who consumed seafood at a higher frequency. This may be linked to the fact that people living near the sea generally have a higher fish consumption compared to inland residents [59,60,61]. It was observed that most people consuming seafood products belong to the age range of >66 years (36.9%), whereas conversely the intermediate (41–65 years old group 53.1%, 26–40 57.3%) and up to 25 years old age groups (56.1%) consume fish much more sporadically.

According to the study by Engle et al. [62] aimed at analyzing the purchasing behavior of pre- and post-COVID-19 fish products, it seems that there has been an increase in the purchase of fish products online exclusively for younger respondents. In addition to the age factor, the level of education seems to have influenced the purchase of products online as with the increase of the level of education emerged a greater tendency to buy fish products online.

About the online market from the analysis of the questionnaire replies, the same trend was observed showing the age and the geographical origin as determining factors for online purchases. For both 2022 and 2024 the age group most likely to buy fish products online is under 25 years of age (71%) coming exclusively from the north (71%)

Indeed, the online market is a growing sector worldwide for seafood; according to a report by the National Fisheries Institute, online seafood sales in the United States reached $1.6 billion in 2020, a 45% increase over the previous year, because consumer demand for fresh, high-quality seafood delivered directly to their homes has grown [63]. The main factors that are boosting online sales of seafood products are the convenience and accessibility of online shopping. Consumers can browse a wide selection of seafood products from the comfort of their homes, comparing prices and quality and placing orders with just a PC or smartphone. Many online seafood retailers also offer free shipping and delivery, further increasing the attractiveness of online seafood shopping. Another key factor is the growing focus on sustainability and transparency in the seafood industry. Many online retailers offer detailed information about the origin and quality of their products, as well as their sustainability practices and certifications. This has helped build consumer trust in online seafood sales [64]. A survey conducted in an area of southern Italy to examine small-scale artisanal fishing showed that about 40 percent of fishermen use smartphones and social media daily for the purpose of marketing products. Digitization, in fact, through apps and websites allows the establishment of a relationship of trust and security between fishermen and consumers. Using online platforms, consumers can in fact, directly reserve freshly caught fish and go directly to the collection point indicated by the fisherman [65]. In Italy, companies such as Pescheria del Sud and ItalMare offer a range of fresh seafood products online, including tuna, swordfish, and octopus. Another successful online seafood marketplace in Italy is Pescheria del Sud, a company specializing in the sale of fresh seafood from southern Italian regions. The company offers a range of products online, including swordfish, tuna, and shrimp, and has experienced significant growth in recent years, with a 20% increase in online sales during the pandemic [66].

The purchase of groceries through online platforms has seen a significant increase during the COVID-19 pandemic, as it was favored by government movement restrictions. Despite this, there are still challenges to be faced regarding this mode of purchase, mainly related to the perishable nature of products and the impossibility for consumers to visually assess the organoleptic qualities of the product [67].

### 3.3. Preferences for Seafood Product

The total questionnaire showed that 44% of respondents usually consumed farmed sea bass. Its consumption is linked to a geographical area, with consumers in the North showing a lower preference for this product (25%). A total of 41% of consumers bought sea bass at large retailers, followed by 36% at fishmongers and 19% at local markets. Furthermore, 78% of consumers usually consumed this product fresh. A total of 92% of consumers responded that it is important to know the traceability of this fish product. In fact, 73% of the respondents recognized that having information on traceability is very important (answering with a score of 5 = very). Regarding the WTP total of the respondents who consume farmed fresh sea bass, the value was 3.097 €/kg, 17.21% higher than the conventional selling price (18 €/kg).

The second product proposed (striped venus clams) was consumed by 51% of the respondents. Residents of Central Italy consumed more striped venus clams than those in the South and North. Regarding the WTP of the respondents who consumed striped venus clams, the value was 2.61 euro/kg; i.e., 26.12% higher than the conventional selling price (10 euro/kg).

The third product proposed in the questionnaire was the giant red shrimp. The questionnaire showed that 41% of respondents habitually consumed this fish product. Its consumption was highest in the South (43%). A total of 38% of consumers bought red prawns at the fishmonger’s, followed by 37% at large retailers and 18% at local markets. About the WTP, for the respondents who consumed red prawns, the value was 6.39 euro/kg, 12.79% higher than the conventional selling price (50 euro/kg).

The last product proposed was albacore tuna. Compared to the other products proposed, 62.3% of the respondents to the questionnaire habitually consumed albacore tuna. Regarding the WTP of the respondents who consumed processed albacore tuna, the value was 4.98 euro/kg, 12.46% higher than the conventional selling price (40 euro/kg).

### 3.4. Statistical Analysis

The Bayes Factors of comparisons with high indications of a statistical relationship are reported in Table 6, Table 7 and Table 8 for 2022, 2024 and full datasets, respectively. The posterior density plots are presented in detail in the Appendix A.

The analyses revealed sex-based differences in consumer behavior. Female consumers were more likely to avoid seafood consumption, and less likely to consume it very frequently (10+ times per month). Males were more likely to buy fresh or processed (but not frozen) seafood from the local market or not buy it at all. Finally, they were highly interested in its traceability, more informed on origin, and would pay a premium for traceable seafood more probably than females for all products, but with the dedicated amount being lower.

Seafood consumption behavior appeared to change with age, with probability of consumption, consumption frequency, preference towards fresh seafood, knowledge of its origin, and interest in its traceability progressively increasing in older age groups. Young consumers (18–25) were the most likely to purchase seafood online, and less likely to purchase it in the fish market or from large retailers. Older consumers (66+) on the other hand, were the most likely to buy seafood from the local market and the most likely to pay a premium for its verified traceability. No clear age trend was evident regarding the source of information on the origin of seafood, except for the high standout tendency of young consumers to be informed by the retailer.

Participants from northern Italy were less likely to consume seafood, while consumption frequency increased across a latitudinal gradient from North to South. As we move from North to South, there was an increased probability of purchasing fresh seafood from the fish- or local markets and being informed by the retailer, while the opposite was true for buying frozen and processed products either from large retailers or online and being informed by ads or the product label. Consumers from the North were the least likely to pay a premium for certified clams and sea bass, while consumers from the South were the most likely to pay a premium for certified shrimp and tuna.

According to the replies, middle school graduates consumed seafood more frequently compared to consumers that had received a higher education, and appeared more interested in its traceability, but were ultimately less informed about its origin. Probability of buying seafood from fish markets and large retailers increased, while online purchasing decreased with higher educational level. University graduates appeared to buy comparatively less fresh seafood. High school graduates were less likely to be informed on seafood origin by ads and the product label. Finally, middle school graduates were more likely to pay a premium price for all of the offered products, but this premium price was lower in the case of shrimp and tuna.

Frequent (10+ times per month) seafood consumers preferred to buy their seafood at the local market, avoided large retailers and were more informed on the product’s origin. Preference towards fresh seafood against frozen and processed increased with consumption rate, as did the WTP a premium for traceable products (except for tuna). However, among the consumers willing to pay a premium, the added price decreasd with more frequent consumption.

Consumers that preferred the fish- and local markets and bought fresh seafood were more informed about seafood’s origin, more interested in its traceability and trusted the retailer more to obtain this information. They were also generally more likely to pay premium prices for traceable seafood (although in the case of tuna this was true independently of the place of purchase or product form). Interestingly, online buyers were more willing to pay a premium for certified shrimp.

Finally, among consumers informed on seafood’s origin and interested in its traceability, the most informed and interested were more likely to pay a premium for different seafood products.

Comparisons per year showed that the ratio of participants consuming seafood dropped in 2024 compared to 2022, but frequency among consumers increased. 2024 participants showed less trust in the retailer for information on seafood origin. WTP a premium for certified shrimp and tuna increased, while the desired quantity dropped in 2024.

### 3.5. Economic Sustainability

The results of the analysis, which highlight the curve of the percentage increase of the BT implementation cost on the average price of fish for each supply chain over the 15 years of the investment duration, are graphically reported in Figure 7, Figure 8, Figure 9 and Figure 10.

In each supply chain graph, the small (S) and large (L) scale curves of the percentage incidence of costs on the base price of the relative product (Pb) are shown. The results of the analysis show the broad sustainability of the application of the new BT in the four production chains, even in reference to the worst cases found for the small-scale scenarios. In fact, the WTP for a certified product by the final consumer is far higher than the percentage increases in the final price due to the implementation of the new BT. For this reason, a break-even point is never reached between the increase in costs due to the implementation of the BT and the increase in the price that the final consumer is WTP for the certified product. This is valid from the first year of investment for all supply chains considered. The largest increases always refer to regional scale scenarios (S), due to the smaller savings obtainable. For the S scenarios, therefore, considering the actualization of all costs in the first year of investment, cost increases are highlighted that vary from +4.4% for the tuna supply chain, to +4.2% for the red shrimp supply chain, which drops to +3.4% for the sea bass supply chain to end with the lowest increase of 2.3% for the clams supply chain.

Figure 11 shows the results of a sensitivity analysis following the application of a percentage change in the market price equal to ±30% for all fish supply chains and for scenarios S and L. The graph highlights more clearly the percentage change in costs as a function of the price change five years after the start of the investment in BT on a large (L) and small (S) scale. In general, the best results (lowest percentages of cost increase, blue area), are referred to the combination of the L scenarios with the highest prices (Ph), on the contrary, the worst results are attributable to the S scenarios in combination with the lowest prices (Pl).

The worst results always refer to the TS with an increase of 1.93% for the S scenario and lowest price (Pl), compared to 0.17% for the L scenario and highest price (Ph). The most competitive is CS with 1.0% in the S × Pl combination, compared to an increase of 0.09% for the L × Ph combination. In an intermediate position compared to the previous ones are the results of RS and BS. The first shows a maximum and minimum increase of 1.84% and 0.16%, respectively, while the second is 1.48% and 0.12%.

There are numerous studies in the literature on the application of BT in the fisheries sector [69]. The excellent potential of BT concerns the ability to uniquely and certainly verify the origin of fishery products and improve traceability in the supply chain, increase transparency, reduce fraud and ensure greater safety of the sector’s products [70]. By applying digital tags and secure RFID and IoT devices along the supply chain, BT can offer certain and immutable control and identification of fishery products from catch to final consumer, avoiding errors and problems in registration and labelling and combating the trade of products resulting from illegal fishing activities and that in some way violate human rights [71].

In the study of Meléndez et al. [46] aimed at analyzing consumers’ WTP a premium for blockchain-certified food products it was found that purchasing behaviors are positively influenced by this technology being driven by informed choices, sustainable and oriented towards ethical food production methods. The WTP an additional surplus in exchange for safer products was also assessed in the study of Dey et al. [47] but specifically for the fish tilapia, pangasius, and rohu using an experimental auction method. The results showed that the WTP is directly proportional to the amount of information provided on production methods and in particular information aimed at ensuring consumer safety.

The evaluation of the economic efficiency of the application of BT can be carried out using many methods, which can be grouped into financial, probabilistic and quantitative [72]. The method adopted in this study was essentially financial and was based on the NPV method, in this specific case, however, applied only to the implementation costs of BT. The economic analysis, built based on the previously exposed cost hypotheses, has highlighted how a possible application of BT for the management of information flows and mandatory registrations for the traceability and tracking of fish from the various supply chains can be economically sustainable. This was particularly true with reference to the orientation of end consumers who, from the specific sector survey conducted, showed a WTP a higher price to have a product traced in a safe and certified manner. The costs of implementing the blockchain showed a strongly decreasing trend as the years of investment duration increased. In the 15th year, i.e., considering the entire investment period, the estimated costs were negligible, showing minimal impacts on the market prices of the respective products in the supply chains. This was also true one year after the start of the investment in BT implementation, at the highest costs. In all cases examined, the costs resulted in a possible percentage increase in the market price of the product that was many times lower than the price increase that consumers had declared to pay for the monitored products. The ratio of the increase in price that the consumer was WTP more to the percentage increase in cost due to BT was at least 11.9 times for CS, 5.3 times for BS and about 3 times for RS and TS.

The present questionnaire showed how important it is for Italian consumers to know the traceability of seafood. In fact, this result is confirmed by other works in the literature. Boncinelli et al. [20] reports how consumers are WTP an average premium of 4.75% to know the catch area of fish used as an ingredient in processed fish foods [20]. As reported by Tamm et al. [25], traceability applied to seafood products allows customers to obtain reliable information about the products they purchase, ensuring certainty of product quality. Specifically, it succeeds in reducing fraud associated with mislabeling or the supply chain, as fishermen and seafood suppliers want their quality products to be traceable by the customer, and at the same time through a traceability system they reduce the possibility of mislabeling of seafood products because they know that by entering the wrong information, the customer will no longer buy the product. In addition, traceability allows the consumer to understand the high price of a quality product. In this way, the producer is able to sell a sustainable product that is not imported from elsewhere but is certified and whose freshness is documented [26]. Therefore, traceability systems have a positive impact on trade in international fish products, demonstrating the consistency and transparency of traceability regulations, thus reducing the likelihood of possible fraud in this area [11]. In addition, the traceability of fishery products can promote sustainable fisheries management, which is unsustainable for the most part worldwide. Through geographical traceability it would be important to demonstrate that a particular fishery product has been legally caught by minimizing fraud and improving fisheries management [4].

Our findings provide an original contribution by demonstrating that, in the Italian fish sector, consumers’ WTP premiums consistently exceed the costs of implementing the blockchain. This empirical evidence not only confirms the economic feasibility of blockchain use, but also highlights its potential to strengthen consumer confidence and support more sustainable consumption practices.

The implementation of blockchain-based traceability systems in the seafood industry presents significant practical challenges, especially in globalized and complex supply chains. Karlsen et al. [73] highlight the need to collect data with a high level of granularity to ensure effective traceability without compromising operational efficiency. Integration with existing IT systems can be difficult due to the variety of platforms used by different actors in the supply chain. Data standardization is an additional obstacle, as formats and procedures differ between countries and companies. Compliance with local and international regulations can also complicate the adoption of the system. Vo et al. [74] emphasize that transaction costs and data governance can be significant barriers to the adoption of the blockchain along the seafood supply chain. Furthermore, Ferreira et al. [75] highlight how managing control processes and creating value through blockchain traceability requires not only advanced technological tools, but also strong coordination between all actors in the supply chain. Stakeholder acceptance and involvement therefore remain critical factors for successful implementation. Blockchain technology in the fishing industry not only poses technological challenges, but also requires adequate governance, data standardization, and collaboration among all actors in the global supply chain.

Although the results of this study indicate that consumers’ willingness to pay exceeds the costs of blockchain implementation, the possible social consequences of even a slight increase in consumer prices cannot be overlooked. In Italy, the introduction of advanced traceability systems, while sustainable from the overall economic point of view, could generate new inequalities in access to certified products if not accompanied by compensatory measures. Consequently, targeted support policies, such as selective subsidies, nutrition education or incentives for the short supply chain, are fundamental to guarantee fairness and inclusiveness, preventing the blockchain from becoming a factor of exclusion rather than strengthening consumer confidence [64].

In conclusion, this work demonstrates the feasibility of implementing the blockchain in the Italian fish sector in terms of economic efficiency, however, to this result we must add the positive effect exerted by BT on the improvement of the traceability and unique identification system of the product and on food safety, producing significant benefits for the final consumer, who has proven to be very sensitive to this issue, showing a high propensity to want to purchase a safe, certified and uniquely traced food product along the entire production chain.

## 4. Conclusions

This work fills an important knowledge gap by demonstrating the economic sustainability of adopting blockchain in the Italian fishing industry. The integrated analysis of implementation costs and WTP premiums provides a solid framework for policymakers and stakeholders, while contributing to the academic debate on the role of digital technologies in promoting sustainable agri-food supply chains.

This study thoroughly demonstrates the economic viability and significant benefits of integrating blockchain technology into Italy’s seafood sector. Beyond just financial gains, the blockchain profoundly improves traceability and ensures unique product identification, which is vital for boosting food safety. Italian consumers are highly responsive to these improvements, showing a strong WTP more for seafood that is safe, certified, and fully traceable from catch to plate. The economic analysis confirms that implementing the blockchain for seafood traceability is sustainable. Consumers’ WTP a premium for certified products far outweighs the increased costs from blockchain adoption, ensuring economic benefits from the very first year. Even with slightly higher initial costs in smaller-scale operations, these become negligible over the investment period, minimally impacting market prices long-term. The impressive ratio of consumer WTP extra versus the cost increase, ranging from 3 to almost 12 times, further highlights its economic appeal.

This research aligns with broader findings on the blockchain’s transformative power in fisheries. It provides reliable information on product quality and origin, reducing fraud from mislabeling and supply chain inconsistencies, and empowers consumers to make informed choices. By ensuring transparency from sea to consumer, the blockchain also helps combat illegal fishing and upholds human rights in the supply chain. This study, consistent with other research, emphasizes that robust traceability systems positively influence international fish trade and significantly contribute to sustainable fisheries management, a critical global issue. Ultimately, this research presents a compelling case for widespread blockchain adoption in the Italian seafood industry, benefiting producers, retailers, and, most importantly, the discerning consumer.

## Figures and Tables

**Figure 1 foods-14-03469-f001:**
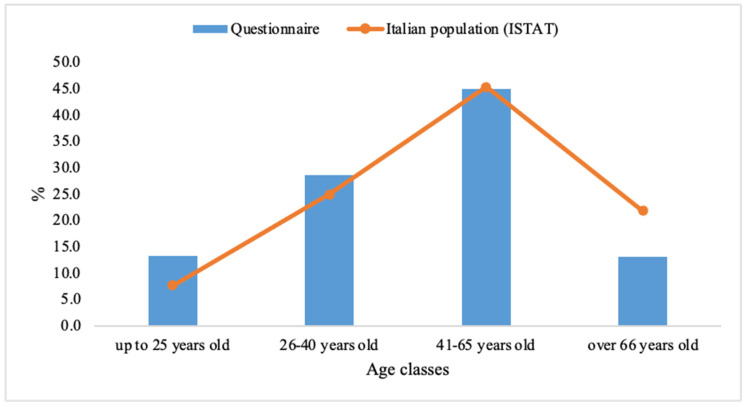
Distribution of the different age groups considered in the questionnaire with respect to ISTAT demographic values.

**Figure 2 foods-14-03469-f002:**
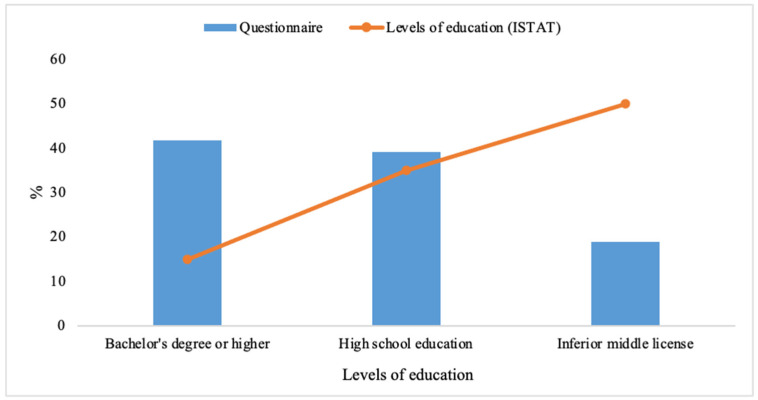
Distribution of the three education levels considered in the questionnaire and comparison with ISTAT education levels.

**Figure 3 foods-14-03469-f003:**
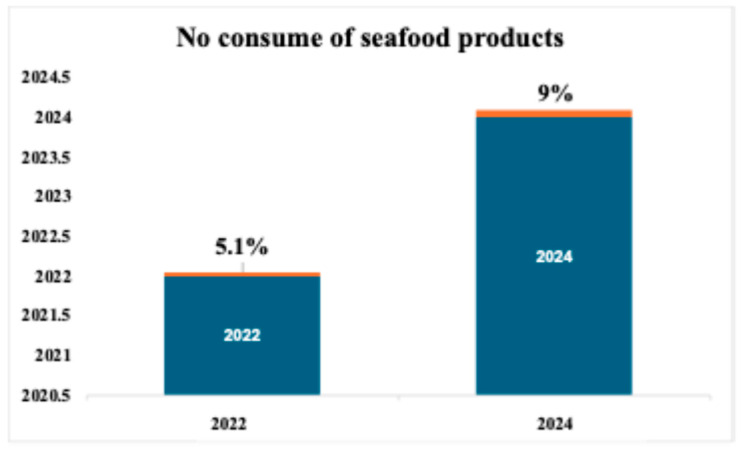
Respondents that declared they do not consume seafood products. The blue bars indicate the estimated percentage of people who reported not consuming fish products in the two years considered. The orange bar represents the margin of error associated with the estimate.

**Figure 4 foods-14-03469-f004:**
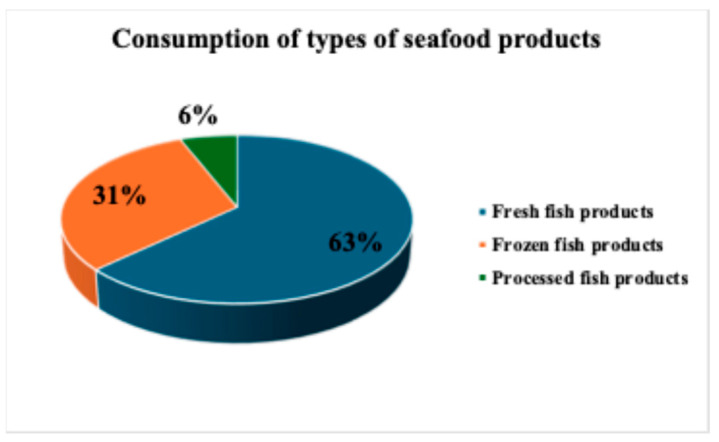
Consumption of types of seafood products.

**Figure 5 foods-14-03469-f005:**
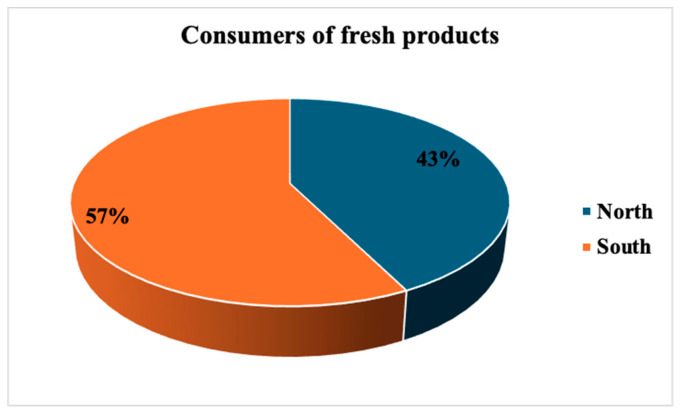
Consumers of fresh products in South and North Italy.

**Figure 6 foods-14-03469-f006:**
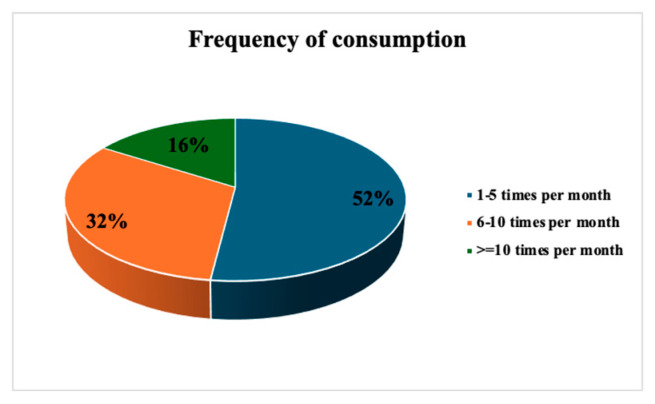
Frequency of consumption of seafood products.

**Figure 7 foods-14-03469-f007:**
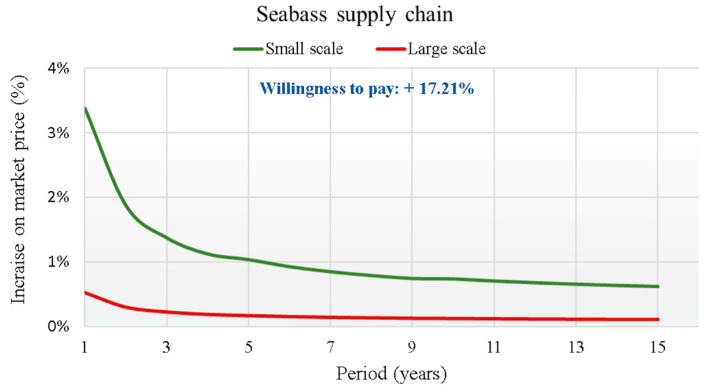
Trend of the cost increase curves on the market price (in €/kg) due to the BT implementation for Seabass supply chain.

**Figure 8 foods-14-03469-f008:**
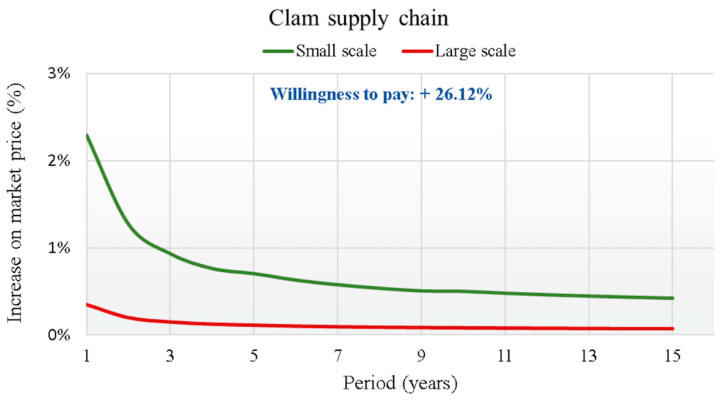
Trend of the cost increase curves on the market price (in €/kg) due to the implementation of the BT for Clam supply chain.

**Figure 9 foods-14-03469-f009:**
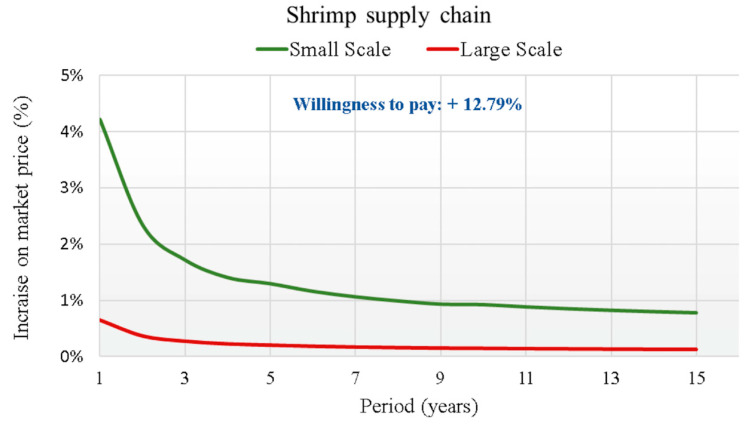
Trend of the cost increase curves on the market price (in €/kg) due to the implementation of the BT for Shrimp supply chain.

**Figure 10 foods-14-03469-f010:**
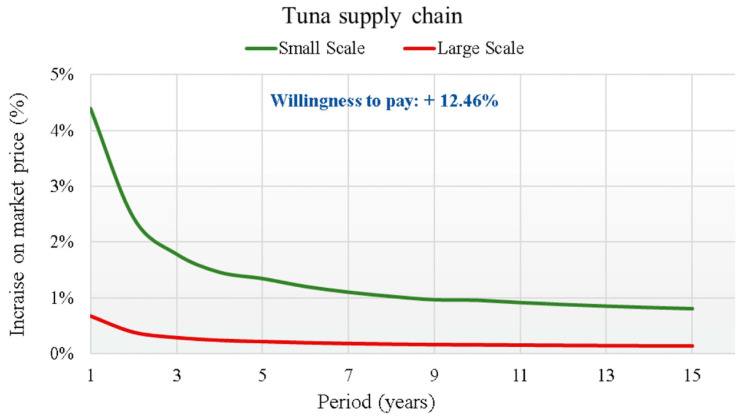
Trend of the cost increase curves on the market price (in €/kg) due to the implementation of the BT for Tuna supply chain.

**Figure 11 foods-14-03469-f011:**
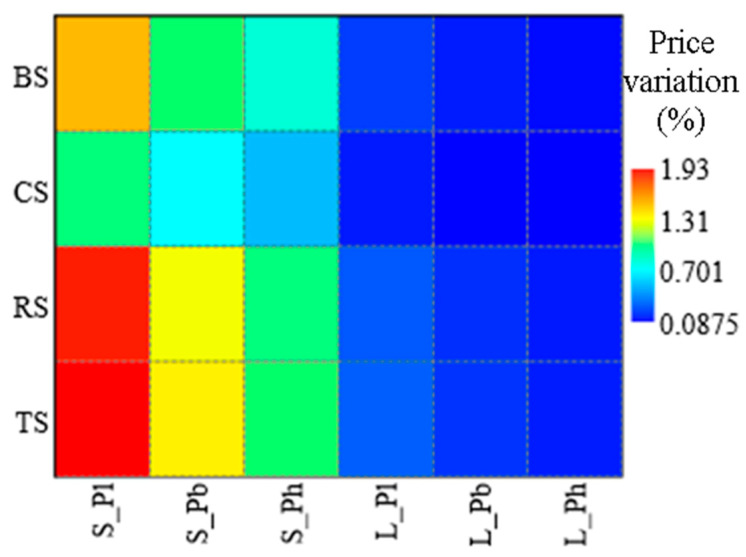
Sensitivity analysis of the price increases at 5th year of investment for implementation of the BT (S-Small Scale; L-Large Scale) for the different supply chains (BS, CS, RS, TS) as a function of a ±30% variation on the base price of the fish (Pl, low price; Pb, base price; Ph, high price).

**Table 1 foods-14-03469-t001:** Questionnaire format implemented in Microsoft Forms.

Index	Questions	Potential Answers
**Q1**	Which is your gender?	Female; Male
**Q2**	Which is your age class?	<25; 26–40; 41–65; >66
**Q3**	Which is your Italian geographical area of residence?	North; Central; South
**Q4**	What is your degree?	>Degree; High school education; inferior middle license
**Q5**	Do you consume seafood products?	Yes; No
**Q6**	How many days a month do you consume seafood products?	Never; 1–5; 6–10; >10
**Q7**	Where do you usually buy seafood products?	Large retailers; local market; fish market; online; do not buy; other
**Q8**	Do you mainly consume fresh, frozen, or processed seafood products?	Fresh; Frozen; Processed
**Q9**	Do you buy seafood products online?	Never; Rarely; Sometimes; Often; Always
**Q10**	Are you informed about the provenance of the seafood you buy?	Yes; No
**Q11**	If yes, how?	Product or point of sale labels; Retailer; Advertisement; Other
**Q12**	Is it important for you to know the traceability of seafood product?	Yes; No
**Q13**	How important is it for you to know the traceability of seafood product?	1 (little); 2; 3; 4; 5 (a lot)
**Q14**	Do you usually consume fresh sea bass raised in Italy?	Yes; No
**Q15**	If traceability of sea bass raised in Italy were guaranteed, how much more would you be willing to spend over the conventional selling price (18 €/kg)?	€
**Q16**	Do you usually consume *striped venus clams*?	Yes; No
**Q17**	If traceability of *striped venus clams* were guaranteed, how much more would you be willing to spend over the conventional selling price (10 €/kg)?	€
**Q18**	Do you usually consume Giant Red Shrimp (*Aristaeomorpha foliacea*—Risso 1827)?	Yes; No
**Q19**	If traceability of Giant Red Shrimp (*Aristaeomorpha foliacea*—Risso 1827) were guaranteed, how much more would you be willing to spend over the conventional selling price (50 €/kg)?	€
**Q20**	Do you usually consume processed *albacore tuna* (canned or jarred)?	Yes; No
**Q21**	If traceability of processed albacore tuna were guaranteed, how much more would you be willing to spend over the conventional selling price (40 €/kg)?	€

**Table 2 foods-14-03469-t002:** Elements considered for the analysis of supply chains in relation to scenarios, quantities produced (year 2023) and variations in market prices (the low and high price were calculated with a variation of ±30% compared to the base price).

Supply Chain	Scenario	Quantity of Fish (Mg y^−1^)	Low Price (€ Mg^−1^)	Base Price (€ Mg^−1^)	High Price (€ Mg^−1^)
**BS**	**Small scale**	422.00	12,600.00	18,000.00	23,400.00
	**Large scale**	6300.00			
**CS**	**Small scale**	1122.00	7000.00	10,000.00	13,000.00
	**Large scale**	16,753.00			
**RS**	**Small scale**	122.00	35,000.00	50,000.00	65,000.00
	**Large scale**	1824.00			
**TS**	**Small scale**	145.86	28,000.00	40,000.00	52,000.00
	**Large scale**	2177.00			

**Table 3 foods-14-03469-t003:** Estimation of economic elements for the calculation of costs of applying the BT in a small- and large-scale scenarios (in k/€).

BT Implementation Phases	Large Scale (L)	Small Scale (S)
**Total initial investment**	500.00	225.00
**BT Network, design, planning, implementation**	250.00	100.00
**Hardware, Infrastructure, Servers, Nodes, Security**	200.00	100.00
**Training, instruction, regulatory adaptation**	50.00	25.00
**Annual maintenance and update of the BT system**	75.00	24.00
**Periodic maintenance and update of the BT (every 5 years)**	100.00	50.00

**Table 4 foods-14-03469-t004:** Purchased seafood in 2022 and 2024 based on geographic area.

2022		
South	Local market	49%
Center	Large retailers	57%
North	Large retailers	63%
**2024**		
South	Local market	60%
South	Fishmongers	51%

**Table 5 foods-14-03469-t005:** The rate of buying frozen and processed seafood increases from the South towards the North both in 2022 and in 2024.

Frozen Seafood		Processed Seafood	
**2022**			
**South**	**North**	**South**	**North**
15%	37%	12%	56%
**2024**			
22%	47%	17%	53%

**Table 6 foods-14-03469-t006:** Bayes Factors of contingency tables and Analyses of Variance (ANOVAs) for the 2022 dataset. Values lower than 3.2 (presented as red dots) are considered “Not Worthy of Mentioning” according to Kass and Raftery [68]. Values in 3.2–10, 10–100 and 100+ constitute “Substantial”, “Strong” and “Decisive” evidence against the null hypothesis of independence, respectively. “-” represents non-defined comparisons.

	Market	Processing	Origin (Y/N)	Origin (How)	Traceability (Y/N)	Sea Bass (Y/N)	Clams (Y/N)	Clams (Extra)	Shrimp (Y/N)
**Age**	●	●	2.04 × 10^3^	●	●	●	●	●	●
**Province**	1.39 × 10^9^	1.45 × 10^2^	●	●	●	3.41 × 10^4^	4.04 × 10^4^	●	●
**Studies**	●	●	●	●	●	●	●	20.06	7.97
**Seafood (Frequency)**	●	●	13.80	●	●	●	●	●	●
**Market**	-	-	9.54 × 10^5^	9.07 × 10^44^	●	4.69	●	●	53.84
**Processing**	-	-	1.06 × 10^14^	4.16 × 10^16^	81.43	6.00 × 10^7^	2.13 × 10^2^	●	24.64
**Origin (Y/N)**	-	-	-	-	-	2.98 × 10^7^	7.99	●	●

**Table 7 foods-14-03469-t007:** Bayes Factors of contingency tables and Analyses of Variance (ANOVAs) for the 2024 dataset. Values lower than 3.2 (presented as red dots) are considered “Not Worthy of Mentioning” according to Kass and Raftery [68]. Values in 3.2–10, 10–100 and 100+ constitute “Substantial”, “Strong” and “Decisive” evidence against the null hypothesis of independence, respectively. “-“ represents non-defined comparisons.

	Seafood (Y/N)	Seafood (Frequency)	Market	Processing	Origin (Y/N)	Origin (How)	Traceability (Y/N)	Traceability (Scale)	Sea Bass (Y/N)	Sea Bass (Extra)	Clams (Y/N)	Clams (Extra)	Shrimp (Y/N)	Shrimp (Extra)	Tuna (Y/N)	Tuna (Extra)
**Sex**	9.84 × 10^8^	2.41 × 10^7^	2.13 × 10^4^	1.37 × 10^12^	5.42 × 10^2^	●	●	4.18 × 10^5^	3.17 × 10^5^	4.13 × 10^2^	5.74	●	7.81 × 10^11^	73.27	6.08 × 10^3^	5.35 × 10^5^
**Age**	1.87 × 10^8^	2.59 × 10^29^	4.66 × 10^57^	1.14 × 10^34^	4.88 × 10^31^	90.70	1.26 × 10^8^	2.98 × 10^24^	1.14 × 10^66^	2.74 × 10^3^	3.63 × 10^13^	10.11	1.47 × 10^37^	1.40 × 10^14^	4.89 × 10^23^	5.44 × 10^26^
**Province**	●	5.12 × 10^26^	1.01 × 10^89^	1.68 × 10^18^	●	5.00 × 10^6^	●	●	4.08 × 10^20^	2.66 × 10^2^	1.10 × 10^5^	●	5.93 × 10^46^	●	8.52 × 10^5^	8.03
**Studies**	●	2.18 × 10^16^	6.43 × 10^42^	2.51 × 10^18^	5.27 × 10^6^	●	62.80	1.68 × 10^9^	3.61 × 10^27^	●	4.70 × 10^2^	●	2.09 × 10^41^	3.77 × 10^4^	1.33 × 10^12^	3.20 × 10^10^
**Seafood (Frequency)**	-	-	1.51 × 10^14^	5.58 × 10^9^	1.50 × 10^6^	●	●	●	1.29 × 10^10^	●	9.98 × 10^3^	●	1.42 × 10^24^	13.54	●	1.20 × 10^10^
**Market**	-	-	-	-	4.19 × 10^15^	1.23 × 10^65^	●	●	9.21 × 10^25^	46.75	5.82 × 10^7^	●	7.13 × 10^28^	3.42 × 10^4^	1.11 × 10^4^	8.73 × 10^11^
**Processing**	-	-	-	-	2.15 × 10^31^	4.82 × 10^35^	4.39 × 10^2^	7.33 × 10^6^	1.98 × 10^38^	●	2.88 × 10^17^	●	2.91 × 10^29^	●	●	1.59 × 10^5^
**Online (Frequency)**	-	-	-	-	●	-	●	●	●	●	●	1.09 × 10^2^	3.28 × 10^2^	●	●	●
**Origin (Y/N)**	-	-	-	-	-	-	-	-	1.64 × 10^5^	●	4.05 × 10^3^	●	3.39 × 10^3^	●	●	5.82
**Origin (How)**	-	-	-	-	-	-	-	-	●	●	●	●	●	●	●	1.95 × 10^2^
**Traceability (Scale)**	-	-	-	-	-	-	-	-	1.82 × 10^6^	●	3.51	●	11.99	●	●	●

**Table 8 foods-14-03469-t008:** Bayes Factors of contingency tables and Analyses of Variance (ANOVAs) for the full dataset. Values lower than 3.2 (presented as red dots) are considered “Not Worthy of Mentioning” according to Kass and Raftery [68]. Values in 3.2–10, 10–100 and 100+ constitute “Substantial”, “Strong” and “Decisive” evidence against the null hypothesis of independence, respectively. “-“ represents non-defined comparisons.

	Seafood (Y/N)	Seafood (Frequency)	Market	Processing	Origin (Y/N)	Origin (How)	Traceability (Y/N)	Traceability (Scale)	Sea Bass (Y/N)	Sea Bass (Extra)	Clams (Y/N)	Clams (Extra)	Shrimp (Y/N)	Shrimp (Extra)	Tuna (Y/N)	Tuna (Extra)
**Year**	1.57 × 10^3^	22.95	●	●	●	9.38 × 10^3^	●	●	●	2.47 × 10^2^	●	20.09	46.53	2.14 × 10^10^	4.93	2.03 × 10^8^
**Sex**	5.40 × 10^8^	3.00 × 10^3^	5.37 × 10^2^	6.18 × 10^10^	13.87	●	●	3.74 × 10^4^	40.59	●	5.79	●	1.26 × 10^10^	73.59	1.03 × 10^3^	91.53
**Age**	4.62 × 10^9^	1.30 × 10^28^	1.23 × 10^60^	1.71 × 10^27^	2.11 × 10^35^	6.59	1.14 × 10^9^	2.02 × 10^25^	1.18 × 10^59^	3.21	4.80 × 10^9^	●	2.77 × 10^30^	1.08 × 10^7^	8.04 × 10^22^	7.51 × 10^15^
**Province**	20.80	8.98 × 10^32^	1.47 × 10^99^	3.08 × 10^21^	●	6.16 × 10^8^	●	●	1.49 × 10^25^	●	4.17 × 10^8^	●	1.69 × 10^44^	●	4.94 × 10^4^	●
**Studies**	●	4.50 × 10^16^	5.38 × 10^47^	1.71 × 10^16^	6.34 × 10^3^	22.95	●	4.25 × 10^5^	7.10 × 10^23^	●	1.55 × 10^3^	●	2.19 × 10^44^	7.42 × 10^4^	4.12 × 10^13^	7.33 × 10^7^
**Seafood (Frequency)**	-	-	3.36 × 10^11^	1.81 × 10^9^	2.80 × 10^10^	●	●	●	8.14 × 10^9^	●	3.61 × 10^6^	●	5.02 × 10^24^	31.74	●	3.30 × 10^5^
**Market**	-	-	-	-	8.83 × 10^23^	1.01 × 10^117^	2.07 × 10^3^	●	3.93 × 10^28^	20.79	4.87 × 10^7^	●	3.16 × 10^31^	2.14 × 10^2^	1.42 × 10^3^	8.64 × 10^4^
**Processing**	-	-	-	-	6.80 × 10^46^	8.81 × 10^57^	2.40 × 10^6^	7.27 × 10^12^	5.30 × 10^46^	●	3.12 × 10^20^	●	1.32 × 10^29^	7.75	26.25	23.75
**Online (Frequency)**	-	-	-	-	●	-	●	●	●	●	●	●	7.90 × 10^3^	●	●	●
**Origin (Y/N)**	-	-	-	-	-	-	-	-	4.05 × 10^12^	●	2.06 × 10^5^	●	5.64 × 10^3^	●	●	●
**Origin (How)**	-	-	-	-	-	-	-	-	14.26	●	●	●	●	●	●	16.90
**Traceability (Scale)**	-	-	-	-	-	-	-	-	1.16 × 10^8^	●	●	●	●	●	●	●

## Data Availability

The original contributions presented in this study are included in the article/Appendix A. Further inquiries can be directed to the corresponding author.

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
