# Peer review of "From Sea to Table: The Role of Traceability in Italian Seafood Consumption"

_foods, 2025, doi:10.3390/foods14203469_

Round 1
Reviewer 1 Report
Comments and Suggestions for Authors
Review of the paper “From Sea to Table: The Role of Traceability in Italian Seafood Consumption” submitted to Foods
The manuscript presents an analysis of a relevant and current issue of interest to the readers of Foods: the traceability of seafood. However, there are many concerns on the way the topic was addressed, with several theorical and methodological weaknesses. This thus challenges the results obtained and the relevance of the conclusions presented. Additionally, the paper lacks coherence and readability in several parts. For that reason, I would not recommend this work for publication in this journal.
Some of the main concerns are briefly summarized below.
- The article lacks a strong conceptual framework linking economic concepts and models related to consumer behaviour, WTP and market failures, like imperfect information – for example, experience goods, credence goods, …. The authors acknowledge that the product they are considering suffers from this (“seafood products are hard to identify and trace and being the most traded food products in the world, they are also the most prone to labeling errors and international frauds”) but do not frame the problem within the appropriate literature of economic science. This weakens the theoretical grounding of the study and limits the explanatory power of the findings.
- In the same line, the economic sustainability assessment intended for the use of BT is not properly supported in theoretical terms in the paper. Additionally, the manuscript does not convincingly justify the purpose of the work, the research gap existent and the contribution to the advancement of knowledge.
- The methodology used and described lack rigour, which jeopardizes the reliability and validity of the results. Some concerns related with:
- The clarity of questions used in the questionnaire. The term “traceability” may be understood differently by different subjects, besides including several aspects, as presented in the Introduction (species identification; production methods; geographic origin..). Therefore, simply asking about the importance of traceability, without presenting a definition or asking subjects if they know its meaning, is problematic.
- It is not clear how credible the estimates presented are for the costs of implementing BT (table 3), as no assumptions and information sources are discussed in the text.
- Except for sample size, sampling methods and techniques (random versus non-random, etc.), and its consequences for extrapolation of results, are not appropriately discussed.
- No mention about the possibility (or not) of subjects responding to the questionnaire twice (in 2022 and in 2024), nor any discussion about the potential implications of additional reflection on the topic – for subjects who had contact with the study / the questionnaire.
- Statistical methods described (section 2.2) without proper clarification of the research hypothesis of the study (“Bayesian approach was followed to provide full distributions for the estimations of the λ parameter, while at the same time allowing for the quantification of the support in favor of the null hypotheses…” – which were the null hypothesis?).
- Lack of clarity in several descriptions and options. For instance, the authors state: “This type of analysis was reported only for the period 2024 because it performs better and because it is more meaningful.” Why is that?
- Secção 3.1. Methodological remarks should not be part of the Results section but part of the section 2. “Materials and Methods”. The points raised are not remarks but important aspects that must be included when presenting the implementation methods of the questionnaire.
- The survey results are poorly presented, relying on long, dense paragraphs instead of clear thematic organization and visual aids such as tables. The information would be much more accessible if structured thematically and supported by tabular data.
- Given that the implementation costs of BT for seafood traceability are unreliable / insufficiently justified in the manuscript, the calculated break-even point and, consequently, the conclusion that “this work demonstrates the feasibility of implementing blockchain in the Italian fish sector in terms of economic efficiency” lacks credibility.
Author Response
Reviewer #1
Review of the paper “From Sea to Table: The Role of Traceability in Italian Seafood Consumption” submitted to Foods
The manuscript presents an analysis of a relevant and current issue of interest to the readers of Foods: the traceability of seafood. However, there are many concerns on the way the topic was addressed, with several theorical and methodological weaknesses. This thus challenges the results obtained and the relevance of the conclusions presented. Additionally, the paper lacks coherence and readability in several parts. For that reason, I would not recommend this work for publication in this journal.
Some of the main concerns are briefly summarized below.
We thank the reviewer for recognizing the importance and interest of the topic, as well as for identifying the weaker points of the manuscript. We addressed the issues they raised, and we believe that in the process we have substantially improved the quality of the manuscript to meet expectations.
The article lacks a strong conceptual framework linking economic concepts and models related to consumer behaviour, WTP and market failures, like imperfect information – for example, experience goods, credence goods, …. The authors acknowledge that the product they are considering suffers from this (“seafood products are hard to identify and trace and being the most traded food products in the world, they are also the most prone to labeling errors and international frauds”) but do not frame the problem within the appropriate literature of economic science. This weakens the theoretical grounding of the study and limits the explanatory power of the findings.
Thank you for your careful observation. We have implemented this in the manuscript.
In the same line, the economic sustainability assessment intended for the use of BT is not properly supported in theoretical terms in the paper. Additionally, the manuscript does not convincingly justify the purpose of the work, the research gap existent and the contribution to the advancement of knowledge.
We thank the reviewer for this valuable comment. We have carefully revised the manuscript to strengthen both the theoretical basis for assessing economic sustainability and the articulation of the scope of the study, the gaps in the research, and the scientific contribution. We have clarified that our economic analysis is based on the net present value (NPV) methodology, a well-established tool for assessing the economic feasibility of technological investments in agri-food supply chains (e.g., Gurtu & Johny, 2019; Francisco & Swanson, 2018). Furthermore, we explicitly state that consumers' willingness to pay (WTP) is used as a measurable proxy for the economic benefit generated by the adoption of blockchain. This approach is consistent with previous studies on blockchain traceability in agri-food systems (Meléndez et al., 2025; Dey et al., 2024). These clarifications have been added in section 2.3.1 (“Description of the economic model”). To better highlight the purpose and originality of the study, we have revised the Introduction and Discussion sections. We now clearly explain that, while blockchain has been studied in other agri-food sectors, there is a lack of systematic analysis of its economic sustainability in the Italian fishing industry. Our study fills this gap by combining cost analysis with consumer WTP at the regional and national levels for several representative fish species. Finally, in the Conclusions, we explicitly state the added value of the study in promoting knowledge about the digitization and sustainability of seafood supply chains.
The methodology used and described lack rigour, which jeopardizes the reliability and validity of the results. Some concerns related with:
The clarity of questions used in the questionnaire. The term “traceability” may be understood differently by different subjects, besides including several aspects, as presented in the Introduction (species identification; production methods; geographic origin..). Therefore, simply asking about the importance of traceability, without presenting a definition or asking subjects if they know its meaning, is problematic.
We added this part “Within the questionnaire, a short video in Italian (https://www.youtube.com/watch?v=j84DAlQNMbw) was presented in order to explain what is traceability and why it is important” in the Materials and Methods section.
It is not clear how credible the estimates presented are for the costs of implementing BT (table 3), as no assumptions and information sources are discussed in the text.
Table 3 was compiled based on previous experience within the PESCA-CHAIN PROJECT (a project studying the economic and technological feasibility of introducing innovative traceability technologies in the fishing industry) and with the help of the AWS pricing calculator (https://calculator.aws/#/createCalculator/ManagedBlockchain). The economic estimate of BT was also performed based on the work of Violino et al. (2019).
Except for sample size, sampling methods and techniques (random versus non-random, etc.), and its consequences for extrapolation of results, are not appropriately discussed.
We are dealing with these concerns in the “Methodological remarks” section, which has now been moved to Materials and Methods from the Discussion, as suggested below by the reviewer. In particular, the following new segments have been introduced to clarify our decisions and their potential implications:
“Not following a stratified approach can create underrepresentation of certain demographics or other groups, leading to potential discrepancies when pooling the outcomes of the survey. The results should therefore be interpreted as rather exploratory, instead of representative of an entire nation. However, we took measures to minimize this potential bias and increase the confidence to the results, which we list below.”
“…To this end, we adapted the analysis to a Poisson sampling scheme (i.e., open sample size, row and column totals), instead of, for instance, joint multinomial sampling (i.e., only sample size is fixed), independent multinomial sampling (i.e., sample size and one of row or column totals are fixed), or hypergeometric sampling (i.e., sample size and both row and column totals are fixed). That decision of sampling scheme affects the way Bayes Factors (see next section) are calculated, and thus, the reported significance. At the same time, following a Bayesian approach based on simulating the data 104 times helped smoothen the effect of outliers.”
No mention about the possibility (or not) of subjects responding to the questionnaire twice (in 2022 and in 2024), nor any discussion about the potential implications of additional reflection on the topic – for subjects who had contact with the study / the questionnaire.
We filtered the dataset for subjects that identified as seafood consumers (i.e., responded “Yes”) and were thus included in the analyses. Out of those, only 15 data rows were identical between 2022 and 2024, meaning that, potentially, 15 subjects could have repeated the survey. We considered this minimal compared to the sample size. We don’t believe that a subject that potentially repeated the survey in 2024 but gave different responses the second time should be excluded, as it reflects the change of opinion of the public in general and is discussed when we compare the outcomes of the two years.
Statistical methods described (section 2.2) without proper clarification of the research hypothesis of the study (“Bayesian approach was followed to provide full distributions for the estimations of the λ parameter, while at the same time allowing for the quantification of the support in favor of the null hypotheses…” – which were the null hypothesis?).
The null hypothesis in contingency tables is that the frequencies of the response variables mirror those of the predictors. That is, in a fictional sample of 100 males and 150 females, if 67 males respond “yes” to a question and 33 respond “no” (i.e., ratio of “yes” vs. “no” ≈ 2), the female frequencies should be roughly 100 “yes” and 50 “no”. If the frequencies differ between groups, significance is calculated based on the extent of the differences relative to the sample size (as in a chi-squared test). Here we selected a Bayesian equivalent, but the principle is the same.
This corresponding segment in the text has been modified to “…while at the same time allowing for the quantification of the support in favour of the null hypotheses (i.e., that observed frequencies of the response variable groups are equal to the frequencies of the predictor groups), instead of only against them”.
Lack of clarity in several descriptions and options. For instance, the authors state: “This type of analysis was reported only for the period 2024 because it performs better and because it is more meaningful.” Why is that?
Thank you for your comment. We have edited the text to eliminate any misunderstandings and modified the captions for some of the figures.
Secção 3.1. Methodological remarks should not be part of the Results section but part of the section 2. “Materials and Methods”. The points raised are not remarks but important aspects that must be included when presenting the implementation methods of the questionnaire.
The survey results are poorly presented, relying on long, dense paragraphs instead of clear thematic organization and visual aids such as tables. The information would be much more accessible if structured thematically and supported by tabular data.
We have moved the section “3.1 Methodological remarks” to “2. Materials and Methods” as suggested. The manuscript has been supplemented with figures and tables in the results section so that these are better described.
Given that the implementation costs of BT for seafood traceability are unreliable / insufficiently justified in the manuscript, the calculated break-even point and, consequently, the conclusion that “this work demonstrates the feasibility of implementing blockchain in the Italian fish sector in terms of economic efficiency” lacks credibility.
This point has been explained in the response in the previous comments.
Reviewer 2 Report
Comments and Suggestions for Authors
The manuscript addresses a relevant and timely topic, namely, how enhanced traceability information, facilitated by the integration of blockchain technology (BT), impacts consumers' willingness to pay (WTP). Overall, the manuscript successfully achieves its proposed objective.
I would suggest a few minor revisions for the authors' consideration:
- In lines 389-395, the authors compare their sample to opinion polls to justify its size. However, this comparison is methodologically weak. Rigorous opinion polls are typically stratified by variables such as gender, age, and critically, educational and socioeconomic levels to ensure representativeness—a step not taken in this study, as evidenced by the educational bias highlighted in Figure 2. It is suggested that the authors acknowledge these crucial distinctions. Furthermore, the manuscript should clarify that sample size alone does not guarantee representativeness; smaller, properly sampled groups can be more representative depending on the sampling method. The discussion should also note that a non-probabilistic sample, such as the one used, does not allow for the calculation of formal confidence intervals for making inferences about the general population.
- While the authors commendably acknowledge the sample's educational bias, the discussion could be strengthened by further reflecting on its implications. The combination of this bias with the absence of socioeconomic indicators (such as income level) in the questionnaire strongly suggests that the respondents likely have a higher-than-average WTP. This could be artificially inflating the premium that consumers are reported to be willing to pay. The authors should discuss this limitation more explicitly. Additionally, it would be valuable to include a brief reflection on the potential societal consequences, such as how increasing the price of seafood through this technology might exacerbate inequalities in access to these products.
- The introduction (lines 63-64) notes that two-thirds of Italy's national demand for seafood is met by imports, particularly from developing countries. Given this, the discussion could be enhanced by briefly addressing the practical challenges of implementing the proposed BT traceability system within these highly globalized and complex supply chains.
- A minor editing note: lines 543-548 appear to be a repetition of lines 505-510.
Author Response
Reviewer #2
The manuscript addresses a relevant and timely topic, namely, how enhanced traceability information, facilitated by the integration of blockchain technology (BT), impacts consumers' willingness to pay (WTP). Overall, the manuscript successfully achieves its proposed objective. I would suggest a few minor revisions for the authors' consideration:
Thank you for your comments.
In lines 389-395, the authors compare their sample to opinion polls to justify its size. However, this comparison is methodologically weak. Rigorous opinion polls are typically stratified by variables such as gender, age, and critically, educational and socioeconomic levels to ensure representativeness—a step not taken in this study, as evidenced by the educational bias highlighted in Figure 2. It is suggested that the authors acknowledge these crucial distinctions. Furthermore, the manuscript should clarify that sample size alone does not guarantee representativeness; smaller, properly sampled groups can be more representative depending on the sampling method. The discussion should also note that a non-probabilistic sample, such as the one used, does not allow for the calculation of formal confidence intervals for making inferences about the general population.
We agree that the comparison with opinion polls is not 100% accurate, since such polls are based on stratified sampling procedures that ensure representativeness with respect to key demographic and socioeconomic variables (e.g., sex, age, education level). Our study, instead, relied on an open online questionnaire, which inevitably introduced biases particularly in terms of education, as also shown in Figure 2. We have now clarified this point in the manuscript by explicitly acknowledging that our sample is not representative of the entire Italian population, but rather of the segment of consumers who are more prone to engage with seafood-related surveys and more familiar with digital tools. We also stress that our findings should therefore be interpreted with caution, as exploratory insights rather than as nationally representative estimates.
We have inserted additional segments in the corresponding part of the manuscript:
“Not following a stratified approach can create underrepresentation of certain demographics or other groups, leading to potential discrepancies when pooling the outcomes of the survey. The results should therefore be interpreted as rather exploratory, instead of representative of an entire nation. However, we took measures to minimize this potential bias and increase the confidence to the results, which we list below.”
“…To this end, we adapted the analysis to a Poisson sampling scheme (i.e., open sample size, row and column totals), instead of, for instance, joint multinomial sampling (i.e., only sample size is fixed), independent multinomial sampling (i.e., sample size and one of row or column totals are fixed), or hypergeometric sampling (i.e., sample size and both row and column totals are fixed). That decision of sampling scheme affects the way Bayes Factors (see next section) are calculated, and thus, the reported significance. At the same time, following a Bayesian approach based on simulating the data 104 times helped smoothen the effect of outliers.”
“…Thus, we do not intend to represent the entire Italian population, but the part of it that consumes seafood (or at least is not against it in principle) and is more willing or prone to participating in online surveys. As such, the results represent an exploratory framework.”
While the authors commendably acknowledge the sample's educational bias, the discussion could be strengthened by further reflecting on its implications. The combination of this bias with the absence of socioeconomic indicators (such as income level) in the questionnaire strongly suggests that the respondents likely have a higher-than-average WTP. This could be artificially inflating the premium that consumers are reported to be willing to pay. The authors should discuss this limitation more explicitly.
We agree that sample size alone does not ensure representativeness and that non-probabilistic sampling limits the possibility of inferring confidence intervals for the general population. We have now clarified this point in the manuscript (“Methodological remarks” section, see answers above). It has to be noted that this is expected to minimally affect the Bayesian contingency table and ANOVA analyses. However, the educational bias and the absence of socioeconomic indicators (such as income level) may have led to an overestimation of the WTP in some cases (but not all of them), which is now included in the discussion.
“It is important to emphasize that the non-probabilistic nature of an open online survey does not guarantee representativeness of the general population, even with a big sample size. This is not expected to have a big effect on contingency table and ANOVA analyses, as they acknowledge the differences in group sizes. However, it can limit the formalization of confidence intervals, the generalization of the results to a nationwide level. Furthermore, it can potentially overestimate the NPV outcome, under the assumption of the higher proportion of higher educational levels in the survey sample corresponds to an indirect overrepresentation of higher incomes, leading to greater WTPs. However, the Bayesian ANOVA showed that studies affected the WTP premium in less than half of the cases (see the results below and the supplementary material for details), and in many of these the effect was negative (i.e., lower educational levels corresponded to greater WTP). Therefore, pinpointing the exact nature and magnitude of this effect is more complicated, and would require further research.”
Additionally, it would be valuable to include a brief reflection on the potential societal consequences, such as how increasing the price of seafood through this technology might exacerbate inequalities in access to these products.
We added this brief reflection as suggested.
The introduction (lines 63-64) notes that two-thirds of Italy's national demand for seafood is met by imports, particularly from developing countries. Given this, the discussion could be enhanced by briefly addressing the practical challenges of implementing the proposed BT traceability system within these highly globalized and complex supply chains.
We implemented this part in the discussion section as suggested.
A minor editing note: lines 543-548 appear to be a repetition of lines 505-510.
As suggested, we deleted the repetition of lines 543-548.
Round 2
Reviewer 1 Report
Comments and Suggestions for Authors
The authors’ efforts to improve the manuscript are acknowledged, resulting in a considerable enhancement of the work. Therefore, I recommend this paper for publication in its current form.
Minor Comments:
-
In Figure 2, replace “1” and “2” with “2022” and “2024.”
-
Lines 957–963 contain a single, overly long sentence. Consider revising it, especially in light of the new sentences introduced immediately before.
-
A careful final proofreading is recommended to correct remaining English typos (for example, Line 254: “It has to be note that” - “It has to be noted that”).